# Exciting Magnetic Dipole Mode of Split-Ring Plasmonic Nano-Resonator by Photonic Crystal Nanocavity

**DOI:** 10.3390/ma14237330

**Published:** 2021-11-30

**Authors:** Yingke Ji, Binbin Wang, Liang Fang, Qiang Zhao, Fajun Xiao, Xuetao Gan

**Affiliations:** 1Key Laboratory of Light Field Manipulation and Information Acquisition, Ministry of Industry and Information Technology, and Shaanxi Key Laboratory of Optical Information Technology, School of Physical Science and Technology, Northwestern Polytechnical University, Xi’an 710129, China; jiyingke@mail.nwpu.edu.cn (Y.J.); wangbinbin@nwpu.edu.cn (B.W.); fangliang@nwpu.edu.cn (L.F.); fjxiao@nwpu.edu.cn (F.X.); 2Qian Xuesen Laboratory of Space Technology, China Academy of Space Technology, Beijing 100094, China

**Keywords:** magnetic dipole mode, plasmonic nanoresonator, photonic crystal nanocavity

## Abstract

On-chip exciting electric modes in individual plasmonic nanostructures are realized widely; nevertheless, the excitation of their magnetic counterparts is seldom reported. Here, we propose a highly efficient on-chip excitation approach of the magnetic dipole mode of an individual split-ring resonator (SRR) by integrating it onto a photonic crystal nanocavity (PCNC). A high excitation efficiency of up to 58% is realized through the resonant coupling between the modes of the SRR and PCNC. A further fine adjustment of the excited magnetic dipole mode is demonstrated by tuning the relative position and twist angle between the SRR and PCNC. Finally, a structure with a photonic crystal waveguide side-coupled with the hybrid SRR–PCNC is illustrated, which could excite the magnetic dipole mode with an in-plane coupling geometry and potentially facilitate the future device application. Our result may open a way for developing chip-integrated photonic devices employing a magnetic field component in the optical field.

## 1. Introduction

Tailoring lightwaves at the nanoscale has opened up new opportunities for effectively controlling light–matter interaction in the applications of surface-enhanced Raman scattering (SERS) [1], fluorescence enhancement [2], nonlinear enhancement [3], localized heating [4], optical trapping [5], and chip-integrated passive [6] and active [7] devices. Plasmonic nanostructures, including metallic nanoparticles, nanoshells, nanorods, nanodimers, bowtie antennas and split-ring resonators (SRRs), provide a route to confining lightwaves to a few nanometers [8]. However, the high-efficient excitation of plasmonic modes in these individual nanostructures is still a challenge, which requires strict mode matches in the parameters of polarization, spatial distribution, wavelength, and wavevector. Conventional free-space excitation technology through an objective lens is inefficient due to the orders of magnitude difference between the spot size of focused light and the cross-section of nanostructures. Moreover, the bulky objective lens is undesirable in on-chip integration applications.

The hybrid plasmonic–photonic structures, the combination of plasmonic nanostructures with photonic waveguides and cavities, provide an approach for the effective excitation of plasmonic modes via the near-field coupling. This also shows many attractive performances on the chip-integrated platform. For example, gold nanostructures were placed on photonic crystal to achieve a hybrid plasmonic–photonic cavity with a high quality (*Q*) factor [9,10], which enables ultra-compact lasers [11], high-throughput biosensors [12], huge electric field enhancement [13], and highly efficient SERS tips [14]. A plasmonic nanodimer was integrated with a microdisk to support chiral emission [15]. Individual plasmonic nanoparticles were combined with a photonic ridge waveguide for sensitive sensing [16], with a photonic micro-ring resonator for the highly efficient excitation of a plasmonic nano-resonator mode [17]. Nevertheless, all of them only related to the electric modes of those plasmonic nanostructures; a CMOS-compatible on-chip excitation approach of their magnetic mode is still missing.

As is well known, the photonic crystal (PhC) can be also used for the design of the slotted PhC waveguides and PhC cavity with ultra-high Q and ultra-small Vm [18,19,20,21,22,23,24,25,26]. Here, the high photon locality of the gold nanostructure and high *Q* of PhC are combined in an optimal way to propose a robust, reliable, alignment-free, and CMOS-compatible approach to effectively excite the magnetic dipole (MD) mode of an individual SRR by integrating it on a silicon planar photonic crystal nanocavity (PCNC). Metal SRR, as a special plasmonic nanostructure, is widely used in metamaterials as a basic MD due to its magnetic dipole-like mode [27,28,29]. Here, SRR is chosen as the metal nanostructure to be integrated on the PCNC surface because it has a similar field distribution profile as that of the PCNC, as demonstrated below. The strong polarization dependence of SRR also enables us to realize light field manipulation from near- to far-field. In addition, the ultra-small mode volume together with strong magnetic field enhancement of the PCNC provides a desirable platform for exciting the magnetic mode of the individual SRR. The mode simulation of the hybrid SRR–PCNC indicates the two nano-resonators have very strong mode coupling with an efficiency exceeding 58%. The excited MD mode can be controlled through the gap width of SRR: the relative position and angle between PCNC and SRR. We further illustrate a structure of a photonic crystal waveguide side-coupled with the hybrid SRR–PCNC structure, which could excite the MD mode with an in-plane coupling geometry and potentially facilitate the future device application. The proposed hybrid SRR–PCNC structure opens an avenue to develop magnetic mode-based devices, such as magnetic sensors [30], magnetic nonlinearity [31], and magneto-optic modulation. Furthermore, field enhancement between the SRR–PCNC gap layer promises the improved performance of silicon integrated electro-optic modulators, photodetectors, and sensors.

## 2. Resonant Characteristics of Individual SRR and PCNC

To analyze and understand the coupling between SRR and PCNC, their resonant characteristics are first studied through scattering spectra of individual SRR and PCNC as well as their magnetic field distribution.

Figure 1a shows the schematic of the employed U-shaped gold SRR with outer lengths of *L*_1_ = 250 nm and *L*_2_ = 210 nm, inner length of *g_y_* = 125 nm, and gap width *g_x_* = 125 nm. The thickness of gold SRR is set as 50 nm. These parameters are optimized to obtain an MD mode of the SRR around the telecom band. The resonant mode of the SRR is simulated by the three-dimensional finite element method (COMSOL Multiphysics). The SRR is set in the air surrounding. For the accuracy of the calculations, the maximum grid size of the model is not larger than one sixth of the light wavelength. The outer boundaries are set as a perfect match layer (PML) for eliminating the interference of the boundary reflection.

With a dipole source excitation located in the center of the SRR, the scattering spectrum of the individual SRR shows two resonant peaks, as shown in Figure 1b, which are the electric quadrupole mode (EQ) and MD mode at the shorter and longer resonant wavelengths, respectively [32]. Compared with dielectric nanocavity, the SRR nanocavity has a lower *Q* of 5.1 and 3.2 for the EQ and MD mode when the *g_x_* = 125 nm, and their value of *Q* will decrease further as the SRR size increases. Both resonant peaks have a large linewidth and low *Q* factor due to the strong radiation of dipolar modes and the Ohmic loss from metal, but this in turn facilitates its coupling with the integrated PCNC, as discussed below. The electric fields of the MD mode at the wavelength of 1430 nm are strongly localized and confined at the ends of SRR’s two arms, as shown in Figure 1c1,c2. The induced circular electrical current indicated by white arrows give rise to a typical MD dipole resonance in the gap region of SRR. As shown in Figure 1c3,c4, the magnetic field *H*_z_ component is concentrated perfectly in the center of SRR, which reveals that the direction of dipole moment is perpendicular to the current plane, i.e., the SRR plane.

The PCNC employed to integrate the SRR is a D-type, which has an ultra-small mode volume and a high *Q* factor, as shown in Figure 1d. The cavity is formed by cutting two inner adjacent air holes into a D shape with a width of D’ = 0.8 D in a hexagonal photonic crystal lattice, where D is the optimized diameter of the air hole [33]. The PCNC is constructed in a 220 nm thick silicon slab, and the lattice constant of the air holes is 454 nm, while the air-hole radius is 127 nm. PCNCs have been widely studied and designed with different defect structures. To evaluate the employee D-shape PCNC, we show a brief comparison of their *Q* and *V*_m_ values in Table 1. The optimized point-defect PCNC has the highest *Q* = 5.02 × 10^6^ and air slot PCNC presents a minimum of *V*_m_ = 0.042 (*λ*/*n*)^3^ [34,35]. The employed D-shape cavity formed by cutting two air holes exhibits a *Q* factor of 10^5^ and *V*_m_ of 0.329 (*λ*/*n*)^3^. Compared with other PC nanocavities, the D-shape cavity has single and concentrated magnetic field distribution, which can more efficiently realize the coupling with the magnetic mode of SRR. Here, because a thin SiO_2_ cladding layer with the air holes being the same as those on the silicon slab is required to integrate the Au SRR on the D-shape PCNC, the asymmetric top and bottom cladding layers cause the decrease in *Q* and increase in *V*_m_.

In the proposed hybrid SRR-PCNC structure, there is a 100 nm thick SiO_2_ spacing layer between them. Consistent with the future device preparation by depositing a SiO_2_ spacing layer on top of the PCNC, the air holes will not be filled by SiO_2_. Hence, the fundamental resonant mode of the D-type PCNC covered with a 100 nm thick SiO_2_ layer is considered here. The obtained resonant wavelength is 1433 nm, as indicated in the scattering spectrum shown in Figure 1e. The resonant peak shows a very narrow linewidth of 0.012 nm, corresponding to a *Q* factor of 1.15 × 10^5^. Figure 1f1,f2 display the magnetic field distribution of the fundamental resonant mode of PCNC at the *x–y* and *x–z* cross-sections, which is strongly localized in the defect region between the two D-type air holes and penetrates the outside of D-type holes. Well-confined mode distribution represents an ultra-small mode volume of *V*_m_ = 0.0342 (*λ*/*n*)^3^. The ultrasmall mode volume together with strong magnetic field enhancement of D-type PCNC promise a desirable platform for exciting the magnetic mode of the individual SRRs through the mode-matching technique.

Figure 2 shows the calculated permeability of gold SRR in the infrared frequency region. Since the operation of the SRR is based on the *LC* resonant circuit (a popular design for the magnetic “atoms” is to mimic a usual *LC* circuit) coupled with a magnetic field, we determine the frequency dispersion of conduction characteristics of metal to fully describe the SRR’s behavior with the help of COMSOL. As shown in Figure 2a, the real part of permeability of SRR is retrieved from the computed scattering and S-para parameter [36,37,38]. In case of *g_x_* = 125 nm, there are two regions with an obvious perturbation of *μ* caused by the EQ resonance and MD resonance of SRR. With the increase in *g_x_*, the magnetic resonance of SRR shifts to the longer wavelength. Thus, we just observe one perturbation of the *μ* region as *g_x_* =250 nm. To emphasize and understand the relationship between *g_x_* and the permeability of SRR, we show the permeability distribution with the changing of *g_x_* in Figure 2b. Compared with permeability, the permittivity of SRR presents a bigger range value and a wider jump region with the changing wavelength shown in Figure 2c,d. Both *μ* and *ε* shift; the perturbation region shifts to the longer wavelength with the increase in *g_x_*. These properties can help us understand the perturbation theory of the electromagnetic field in the paper. Compared with the excitation via incidence of plane wave or dipole source, there are more complicated situations when SRR is excited by PCNC. This is because the angle degree and electric polarization cannot be accurately determined. The exact magnitude of permeability and permittivity cannot be derived when SRR is integrated on the PCNC, but the general trend can be inferred from the above results.

## 3. Magnetic Dipole Mode Excitation: Strong Coupling between SRR and PCNC

Figure 3a schematically displays the hybrid SRR-PCNC structure, which has a 100 nm thick SiO_2_ spacing layer between them. Since the SRR is located at the center of the PCNC, resonant coupling between their modes would be realized. Perturbation of the PCNC resonant mode by the SRR could be observed due to the large effective permittivity and permeability of the resonant modes from the gold SRR. On the other hand, SRR’s MD mode would be effectively excited by the near-field of the PCNC mode.

To this end, we first change the gap width *g_x_* of SRR to study the coupling effect between SRR and PCNC in view of the fact that *g_x_* strongly affects the resonant wavelength of SRR [39]. As indicated in Figure 3b, MD and EQ modes of SRR sequentially appear around the resonant peak of PCNC (at 1433 nm) when *g_x_* is increased gradually. As a result, the scattering spectra of hybrid SRR–PCNC structures show a dynamic change. As *g_x_* = 50 nm, the emission spectrum exhibits a narrow line width, and the resonant wavelength hardly changes. Especially, as the resonance wavelength of the MD or EQ modes of SRR overlap the peak of PCNC, the spectrum of SRR–PCNC presents an obvious blueshift and wider line width when *g_x_* = 125 and 295 nm.

To explain the variation of the scattering spectra of the hybrid SRR–PCNC structure as a function of the SRR gap width of *g_x_*, we plot the resonant wavelengths of MD and EQ modes of individual SRR and the resonant wavelength shift ∆*λ* of the hybrid SRR–PCNC structure relative to the resonant wavelength of PCNC, as shown in Figure 3c. As the SRR gap width *g_x_* increases from 50 to 500 nm gradually, the resonant peaks of the MD mode and EQ mode overlap with the resonant peak of PCNC successively. As a result of their mode couplings, the resonant mode of SRR would induce an effective permeability perturbation *μ_SRR_* and an effective permittivity perturbation *ε_SRR_* to the resonant mode of PCNC, respectively. In turn, this leads to a resonant frequency shift ∆ω or ∆*λ* of PCNC, which is given by a perturbation theory [40,41].
(1)$$\Delta \omega ∼\frac{\omega_{n}}{2}\frac{\delta \mu {\left|H_{n,z}\left(\vec{r_{0}}\right)\right|}^{2}}{∭{}_{-\infty }^{+\infty }r{\vec{H}}_{n}^{*}\cdot {\vec{H}}_{n}\left(\vec{r}\right)dV}$$
where *ω*_n_ is the resonant frequency of PCNC, ${\vec{H}}_{n}(\vec{r})$ and δ and *μ* are the unperturbed magnetic field and the magnetic perturbation, respectively. ${\vec{r}}_{0}$ is the position of SRR. Both the perturbations from permittivity *ε_SRR_* and permeability *μ_SRR_* cause the resonant frequency shift ∆*λ*. The MD mode is getting close to the mode of PCNC at 1433 nm when *g_x_* increases gradually (Figure 1b,c). This induces a permeability perturbation *μ_SRR_* > 0 at the wavelength of 1433 nm, as calculated in Figure 2, and hence leads to a red shift of the resonant wavelength of the hybrid SRR–PCNC structure (Figure 3c). As indicated by Equation (1), *μ_SRR_* decreases to 0 when *g_x_* = 105 nm, whereas a dramatic decrease in ∆*λ* is observed in Figure 3c. When the gap width *g_x_* changed from 105 to 125 nm, *μ_SRR_* remains negative. According to Equation (1), the blue shift of the resonant wavelength of the hybrid SRR–PCNC happens in this case. Especially, when the MD resonant wavelength matches the PCNC mode wavelength at *g_x_* = 125 nm, a maximum inductive magnetic field in the SRR is obtained, as shown in Figure 3d. When the range of *g_x_* changed from 125 to 225 nm, both MD and EQ modes are weak at 1433 nm, and ∆*λ* is nearly flat. With the further increase in gap width (*g_x_* > 225 nm), the EQ mode dominates in SRR and takes over the MD mode. In this case, SRR first presents a negative permittivity *ε_SRR_* and gives rise to an extraordinary blue shift of the resonant wavelength of the hybrid SRR–PCNC. The maximum blue shift happens at *g_x_* = 295 nm, where *ε_SRR_* = 0. As the resonant wavelength of the EQ mode moves away from the PCNC mode when *g_x_* > 295 nm, *ε_SRR_* is positive, which causes a redshift of the resonant wavelength of the SRR–PCNC structure.

To find a suitable *g_x_* to effectively excite the MD mode of SRR, we plot the coupling efficiency *η* and *Q* factor of the hybrid SRR–PCNC structure as a function of the gap width *g_x_* (Figure 3e). Here, the coupling efficiency *η* is calculated by [42].
(2)$$\eta =\frac{\underset{SRR}{∭}\left(\mu_{0}{\left|\vec{H}\right|}^{2}+\varepsilon_{0}{\left|\vec{E}\right|}^{2}\right)dV}{∭{}_{-\infty }^{+\infty }\left(\mu_{0}{\left|\vec{H}\right|}^{2}+\varepsilon_{0}{\left|\vec{E}\right|}^{2}\right)dV}$$
where $\mu_{0}$ and $\varepsilon_{0}$ are the vacuum permeability and vacuum permittivity, $\vec{H}$ and $\vec{E}$ represent the magnetic and electric, respectively.

According to the black curve in Figure 3b, the individual PCNC has a high *Q* factor. Integrating an SRR on PCNC introduces a perturbation to the resonant mode of PCNC, leading to a reduction of *Q* factor with the increase in *g_x_*. For a small *g_x_*, the MD resonant peak of the SRR is far away from the resonant peak of PCNC, as shown by the red curves in Figure 3b, which indicates weak coupling between SRR and PCNC; then, a high *Q* factor inherits from PCNC (Figure 3e). The increased *g_x_* moves the MD resonant peak of SRR close to that of the PCNC (Figure 3b) and hence enhances perturbation to the resonant mode of PCNC, which brings a drastic drop of the *Q* factor of the hybrid SRR–PCNC structure (Figure 3d). The *Q* factor nearly keeps flat if *g_x_* > 100 nm, which is caused by the dramatic growth of loss of the SRR. If we enlarge the view of Figure 3d in the inset, we can clearly observe a fluctuation of *Q* values. A minimum *Q* factor appears at *g_x_* = 105 nm (inset of Figure 3d), which indicates the strongest perturbation of the SRR to PCNC. Close to this position, the maximum coupling efficiency of 58% is obtained at *g_x_* = 125 nm, where the MD resonant peak of the SRR has an overlap with the resonant peak of PCNC. The MD mode of the SRR is effectively excited at *g_x_* = 125 nm (Figure 3e). Another minimum *Q* factor point appears at *g_x_* = 260 nm and the maximum coupling efficiency of 26% is obtained at *g_x_* = 250 nm (inset of Figure 3d). In this case, the EQ resonant peak of the SRR is close to the resonant peak of PCNC instead of the MD peak (Figure 3b). Hence, the EQ mode of the SRR is effectively excited at *g_x_* = 250 nm.

To evaluate the intrinsic properties of the hybrid SRR–PCNC and the strength of coupling between the MD mode of the SRR and the resonant mode of the PCNC, the mode volume *V*_m_ and figure of merit *Q*/*V*_m_ are calculated in Figure 4a,b. The values of *V*_m_ have two dips when the gap width of SRR increases. This is because the MD mode and EQ mode of SRR are excited by the PCNC mode as *g_x_* = 125 nm and 250 nm, which cause the strong confinement of photons. Meanwhile, the values of *Q*/*V*_m_ also have peaks as *g_x_* = 125 nm and 250 nm, which indicate the higher Purcell effect of the SRR–PCNC. The overall declined trend of *Q*/*V*_m_ as the *g_x_* increased is because of the rapid decrease in *Q* factors due to the increased metal loss with the larger SRR. The comparison of coupling efficiency in the hybrid plasmonic–photonic structures is shown in Table 2. Note that in our proposed SRR–PCNC structure, the MD mode is successfully excited with a high coupling efficiency. The coupling efficiency of the hybrid SRR–PCNC can be improved by optimizing the size and position of the SRR to increase the overlapping between the PCNC mode and MD mode of the SRR.

## 4. Effect of Relative Position and Angle between SRR and PCNC

Adjusting the gap width of SRR offers a versatile method to control the excitation of the MD mode. However, it introduces an undesirable EQ mode. In this section, we present the fine controlling of the excitation of the MD mode through the relative position and angle between the SRR and PCNC. The SRR with the fixed *g_x_* = 125 nm is employed to provide the MD mode overlapping with the mode of PCNC.

Figure 5a schematically shows the horizontal distance of SRR, *x_p_*, relative to the center of the PCNC denoted by the arrows. The scattering spectra of the hybrid SRR–PCNC structures exhibit periodic red-shifted and blue-shifted resonant peaks as the SRR moves from left to right gradually, as shown in Figure 5b. It needs to be pointed out that the line width of the resonant spectrum indicates the couple between the SRR and PCNC and loss of the hybrid SRR–PCNC structure. For example, as the *x_p_* = ±*a*, the narrow line width reveals the weak coupling and lower loss due to the scattering and absorption of the SRR. Resonant wavelength shifts with respect to the resonant wavelength of the bare PCNC as the SRR moves are shown in Figure 5c. The change of wavelength red shift and blue shift could be attributed to the weak and strong coupling between the MD mode of the SRR and the PCNC mode with different magnetic field intensity, which is governed by Equation (1). The value of *V*_m_ as the function of the SRR position is also shown with a red line. The minimum value of *V*_m_ indicates the most confinement of photons determined by the coupling strength of the hybrid system. In addition, the different linewidths in Figure 5b indicate the different coupling efficiency between the SRR and PCNC as well, which is extracted as the *Q* factors of the hybrid SRR–PCNC structure, as shown in Figure 5d. Both the coupling efficiency and *Q* factor indicate the strong interaction between the SRR and PCNC when the SRR locates in the position where the PCNC mode has the maximum magnetic field at *x_p_* = 0, ±*a*, and weak interaction when *x_p_* = ±*a*/2 where the magnetic field is close to 0.

Tuning the SRR direction with different relative angle *θ* with respect to the PCNC is another method to finely adjust the MD mode of the SRR, as schematically shown in Figure 6a. An obvious red shift and sharpening of the resonant peak are observed from the scattering spectra of the hybrid SRR–PCNC structure when the relative angle is changed from 0° to 90°, as shown in Figure 6b. These results can be interpreted by the change of coupling efficiency between SRR and PCNC. When the relative angle changes from 0° to 90°, the magnetic field overlapping between the SRR and PCNC will decrease gradually. This means that the coupling efficiency between the SRR and PCNC will be reduced, corresponding to a decrease in the magnetic response and loss. Then, there is a bigger permeability of SRR according to Figure 2, which leads to the sharpening of the peak and red shift of the resonant wavelength.

The resonant wavelength shift Δ*λ* remains negative due to the negative permeability of the SRR and exhibits periodic variation similar to the sinusoidal function of *θ* (Figure 6c). When *θ* = 0°, 180°, and 360°, the resonant wavelength has a maximum blue shift, indicating the strongest coupling between the SRR and PCNC. When *θ* = 90° and 270°, the resonant wavelength shows the minimum blue shift, showing the weak coupling between the SRR and PCNC. The maximum blueshift corresponds to the minimum of *V*_m_, which predicts the strongest confinement of photons as the most coupling between the SRR and PCNC. This is confirmed by the coupling efficiency and *Q* factor curves in Figure 6d. The coupling efficiency (*Q* factor) gradually weakens (increases) when *θ* increases from 0° to 90°, and it enhances (decreases) from 90° to 0°.

## 5. All Integrated On-Chip Excitation of Magnetic Dipole Mode of SRR

To facilitate the future device applications of the near-field excited MD mode of SRR in the hybrid SRR–PCNC structure, we further propose to side-couple the SRR–PCNC structure with a photonic crystal waveguide, enabling the in-plane accessing geometry.

Figure 7a is the schematic to excite the MD mode of the SRR in the integrated architecture. The cavity mode of the PCNC is excited through a photonic crystal line-defect waveguide, which then excites the MD mode of the SRR. By optimizing the mode overlap between the PCNC and the photonic crystal waveguide, which have a separation with three rows of air holes, their evanescent field coupling efficiency could be as high as 78% [42]; as shown by the *x−y* cross-section mode profile in Figure 7b, the resonant mode of D-type PCNC is effectively excited by the photonic crystal waveguide, immediately following effective excitation of the MD mode of the SRR. As shown in the cross-section view of Figure 7c,d, the magnetic field of the resonant mode in the PCNC can effectively couple into that of the SRR. As calculated from Equation (2), the excitation efficiency of the SRR MD mode by the PCNC mode is 58%. Hence, the total excitation efficiency of the SRR MD mode is estimated as 45% from the side-coupled photonic crystal waveguide. Finally, we also calculated the transmission spectra of SRR–PCNC and PCNC, respectively, as shown in the inset of Figure 7d. Corresponding to the resonant characteristics of the PCNC and SRR–PCNC, a broader and blue-shifted resonant dip is obtained from the SRR–PCNC compared with that of the PCNC. These results support the all-integrated on-chip excitation of the MD mode of the SRR, paving the way to applications in chip-integrated magnetic-optic devices.

## 6. Conclusions

In conclusion, we have demonstrated an excitation approach of the MD mode of the SRR by integrating it onto a silicon PCNC, which eliminates the use of a bulky objective lens and significantly improves the efficiency. This benefits from the strong near-field mode coupling between the SRR and PCNC. The coupling mechanism in the hybrid SRR–PCNC structure is numerically and theoretically illustrated with the help of an electromagnetic perturbation theory and finite element method (FEM). A coupling efficiency between the SRR MD mode and PCNC mode as high as 58% indicates the highly efficient on-chip excitation of the SRR MD mode. A fine adjustment of the excitation of the MD mode is presented through the varied relative position and angle between the SRR and PCNC. This result shows the intensity distribution of the near-field in the PCNC and reveals the potential application in magnetic imaging. An all-integrated excitation configuration of the MD mode is finally described by side coupling a photonic crystal waveguide with the PCNC. Due to the limitation of experimental conditions, the above results cannot be verified by experiments for the time being. Therefore, we use the finite difference time domain (FDTD) technique to verify the simulation and find that the results of the two algorithms are consistent. Our approach may open the way for the on-chip applications of a magnetic field component in the optical field, such as a magnetic sensor [30], magneto-optic modulator [31,43,44,45], second harmonic manipulation [46], all-optical switch [47], microwave sensor [48], plasmonic nanolasers, and spacer [49,50,51].

## Figures and Tables

**Figure 1 materials-14-07330-f001:**
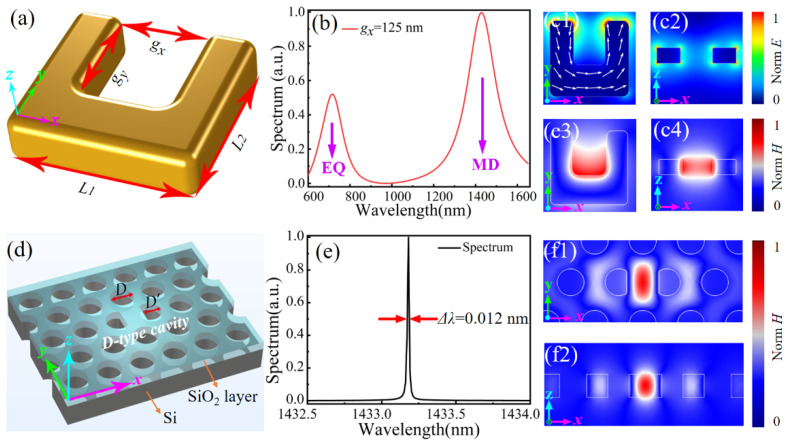
(**a**) Schematic of a gold SRR with defined geometric parameters. (**b**) Calculated scattering spectrum of an individual SRR showing EQ and MD resonant modes. (**c1**,**c2**) Electric field and (**c3**,**c4**) magnetic field distributions of SRR at the resonant wavelength of 1430 nm. (**d**) Schematic of the D-type PCNC with D’ = 0.8 D. (**e**) Calculated scattering spectrum of the D-type PCNC. (**f1**,**f2**) Magnetic field distributions of D-type PCNC at *x−y* and *x−z* cross-sections, respectively.

**Figure 2 materials-14-07330-f002:**
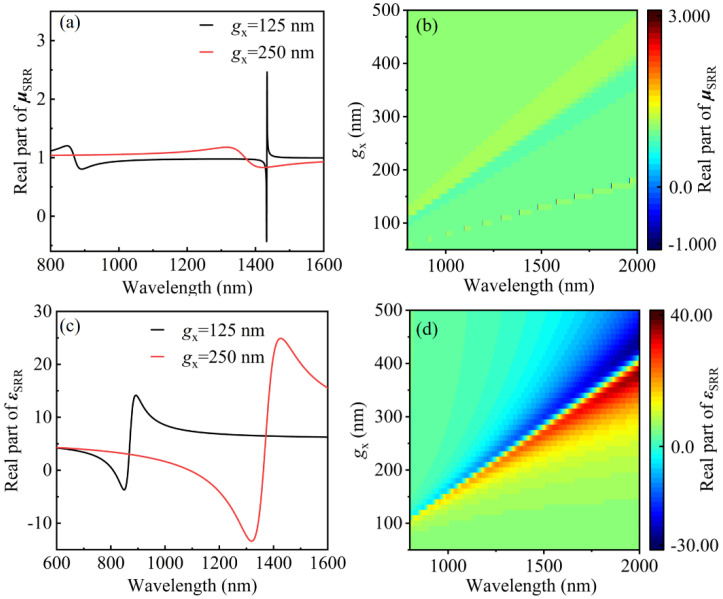
(**a**) The real part of the retrieved effective permeability *μ* around the plasmon resonant of single SRR. (**b**) The *μ* distribution about the *g_x_* size of SRR from 50 to 500 nm. (**c**) The real part of the retrieved effective permittivity *ε* around the plasmon resonant of single SRR. (**d**) The *ε* distribution with the *g_x_* size of SRR from 50 to 500 nm.

**Figure 3 materials-14-07330-f003:**
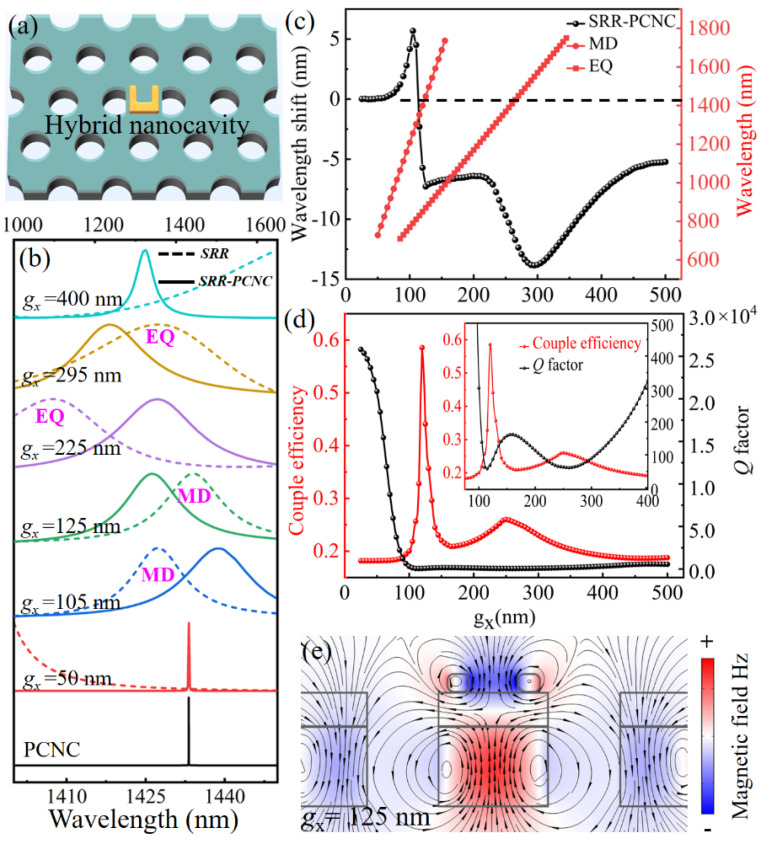
(**a**) Schematic of the proposed hybrid SRR–PCNC structure. (**b**) Scattering spectra of individual SRR and hybrid SRR–PCNC structures as a function of the SRR gap width. (**c**) Resonant wavelength of MD and EQ modes of individual SRR (right), and resonant wavelength shift ∆*λ* of a hybrid SRR–PCNC structure relative to the resonant wavelength of the bare PCNC as functions of the SRR gap width. (**d**) Variations of *Q* factors of hybrid SRR–PCNC structure and coupling efficiency between SRR and PCNC with different SRR gap widths. Inset: Zoomed *Q* factor at the flat region. (**e**) Simulated magnetic field distribution of MD mode of hybrid SRR–PCNC structure (*x–z* cross-section) with the SRR gap width of 125 nm.

**Figure 4 materials-14-07330-f004:**
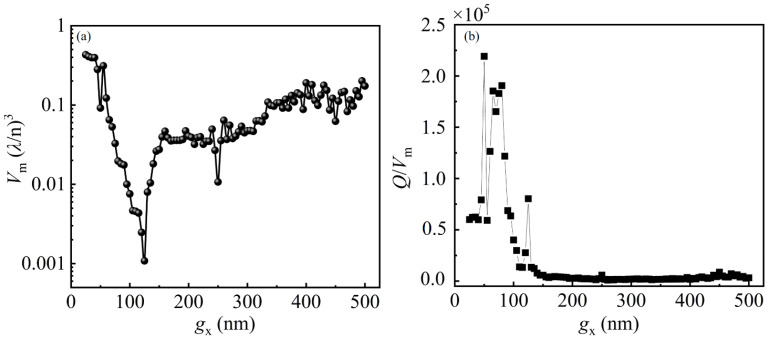
(**a**) The calculated *V*_m_ as a function of *g_x_*. (**b**) The simulated *Q*/*V*_m_ ratio as the function of *g_x_*.

**Figure 5 materials-14-07330-f005:**
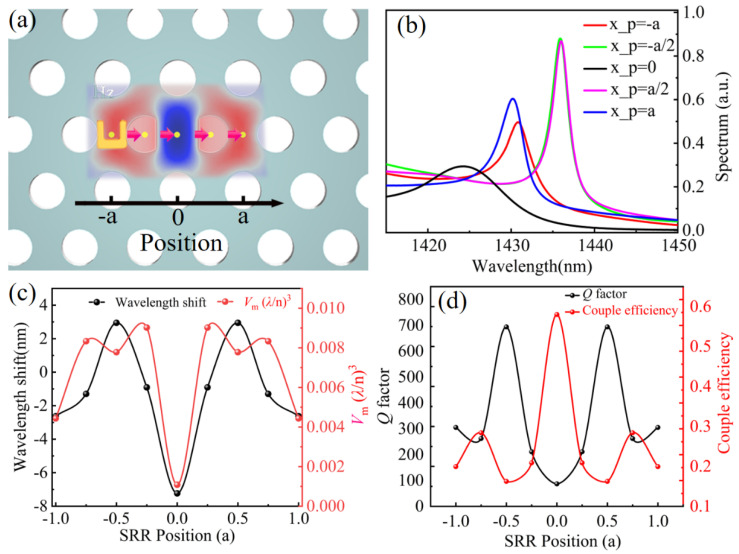
(**a**) Schematic for the changing relative position between the SRR and D-type PCNC. The magnetic field distribution of the mode in PCNC at 1433 nm is superposed onto the structure. (**b**) Scattering spectra of the MD mode of the hybrid SRR–PCNC structure with different relative positions. (**c**) Resonant wavelength shift and value of *V*_m_ under different relative positions. (**d**) *Q* factor and coupling efficiency as a function of the relative position.

**Figure 6 materials-14-07330-f006:**
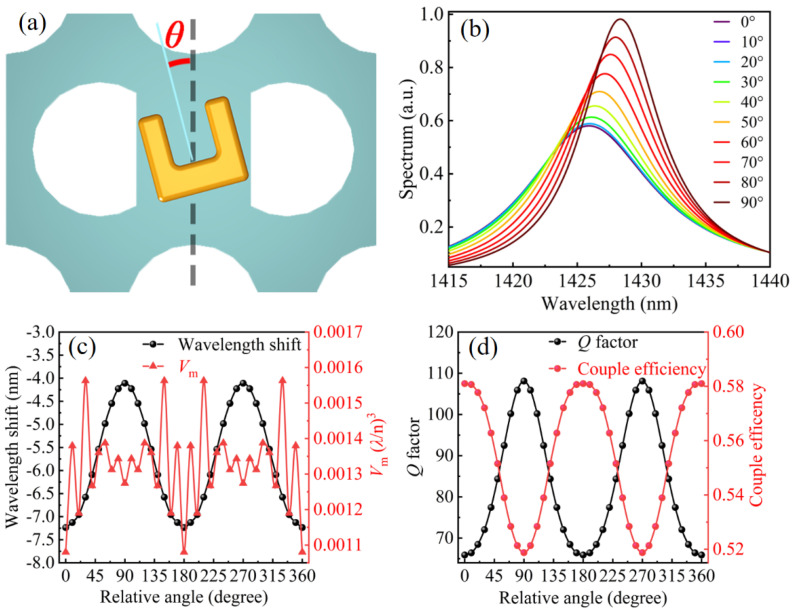
(**a**) Schematic of the turning relative angle *θ* between SRR and PCNC. (**b**) Scattering spectra of hybrid SRR-PCNC structure with different relative angle from 0° to 90°. (**c**) Resonant wavelength shift Δ*λ* and value of *V*_m_ as a function of *θ*. (**d**) Coupling efficiency *η* and Q factor as functions of *θ*.

**Figure 7 materials-14-07330-f007:**
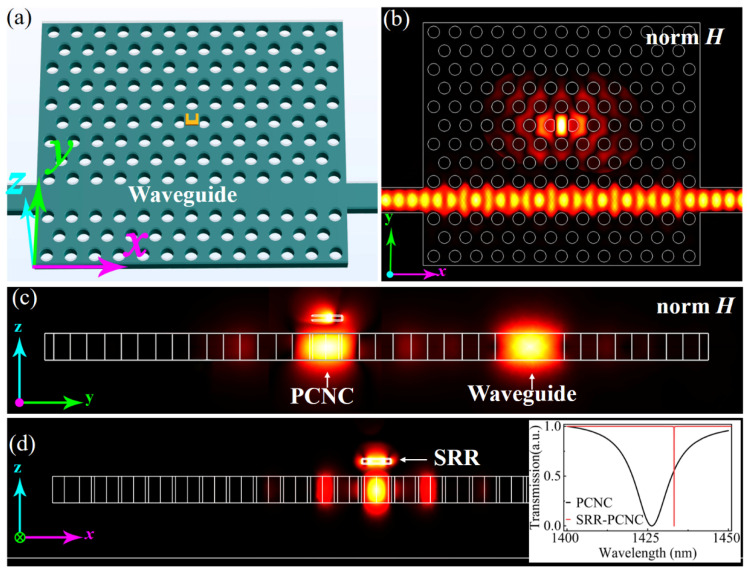
(**a**) Schematic of all integrated on-chip excitation of the MD mode in the SRR by side-coupling the hybrid SRR–PCNC with a photonic crystal waveguide. (**b**–**d**) Simulated magnetic field distribution of the proposed configuration in (**a**) from the views of the *x−y* cross-section through the center of the silicon slab (**b**), *y−z* cross-section (**c**), and *x−z* cross-section through the center of the PCNC (**d**), respectively. The inset is the calculated transmission spectrum of the SRR–PCNC and PCNC side-coupled by a waveguide.

**Table 1 materials-14-07330-t001:** Comparison of different PCNCs.

Structure	Quality Factor (Q)	Mode Volume (*V*_m_)	Ref.
Heterostructure PCNC	4 × 10^4^	1.46 (*λ*/*n*)^3^	[34]
Air slot heterostructure PCNC	2.6 × 10^4^	0.042 (*λ*/*n*)^3^	[35]
Optimized point-defect PCNC	5.02 × 10^6^	0.6 (*λ*/*n*)^3^	[23]
D-shape PCNC	2.005 × 10^5^	0.329 (*λ*/*n*)^3^	[33]

**Table 2 materials-14-07330-t002:** Comparison of coupling efficiency of the hybrid structure.

Structure	*η*	Excitation of MD (Yes or No)	Ref.
Gold spheres dimer on PhC cavity	40%	No	[9]
Gold bowtie antenna on PhC cavity	62%	No	[13]
Gold particle on waveguide	9.7%	No	[16]
Gold particle on ring resonator	78%	No	[17]
Nanowires on PCNC	1.9%	No	[42]
Gold SRR on PCNC	58%	Yes	−

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
