# Peer review of "Exciting Magnetic Dipole Mode of Split-Ring Plasmonic Nano-Resonator by Photonic Crystal Nanocavity"

_materials, 2021, doi:10.3390/ma14237330_

Round 1

Reviewer 1 Report

In the present paper, the authors purpose a high-efficient excitation approach of magnetic dipole
mode of the split-ring resonator (SRR) by photonic crystal nanocavity (PCNC). The authors reported
that high excitation efficiency up to 58% is realized through the resonant coupling between the
modes of the SRR and PCNC. They show that by tuning the relative position and twist angle
between SRR and PCNC, fine adjustment of the exciting magnetic dipole mode is demonstrated. In
addition, a structure with a photonic crystal waveguide side-coupled with the hybrid SRR-PCNC is
studied in the present paper. Such structure could excite the magnetic dipole mode with an in-plane
coupling geometry.
Hence, the study is thorough and this manuscript can serve as a design guideline for silicon
integrated electro-optic modulators, photodetectors, and sensors. I would agree that compared with
other studies on H1 and L3 PhC cavities, the presented work has the advantage of design
simplicity. The manuscript is well organized and their considered subject is noteworthy. However,
the introduction does not give a satisfactory literature survey on a similar topic.
In short, the manuscript is only suitable for publication if the following questions and concerns can
be addressed:
1. Other than numerous instances of poor grammar and a number of sentences without a
verb, the paragraph/sentence describing the obtained results in the presented work is
repeated several times at different places in the manuscript.
2. In the first section of the paper (Introduction), the authors mentioned that, gold nanostructures
were placed on photonic crystal to achieve hybrid plasmonic-photonic cavity with
high quality (Q) factor, which enables ultra-compact laser, high-throughput bio-sensor. In
fact, the characteristics of the PhC can be also used for the design of the slotted PhC
waveguides and PhC cavities with ultra-high Q-factors and ultra-small mode volumes
(Kassa-Baghdouche, L., & Cassan, E. (2018). Photonics and Nanostructures-
Fundamentals and Applications, 28, 32-36, Nakamura, T., Takahashi, Y., Tanaka, Y.,
Asano, T., & Noda, S. 2016 Optics express, 24(9), 9541-9549, Kassa-Baghdouche L,
Boumaza T, Cassan E and Bouchemat M 2015 Optik 126 3467-3471). These references
should be included in the introduction of the paper.
3. In the second section (Resonant Characteristics of Individual SRR and PCNC), the
proposed PhC cavity is formed by cutting two inner adjacent air-holes into D-shape in a
hexagonal photonic crystal lattice, where D is the diameter of the air-hole. However, in the
literature, various point-defect PhC cavities have already been well proposed and analyzed
by many researchers (Kassa-Baghdouche, L. (2020), JOSA B, 37(11), A277-A284.
Nakamura, T., Takahashi, Y., Tanaka, Y., Asano, T., & Noda, S. 2016 Optics express,
24(9), 9541-9549, Kassa-Baghdouche, L., &Cassan, E. (2020). Optical and Quantum
Electronics, 52(5), 1-13). The difference between these structures and the proposed
structure should be included in the paper. Moreover, such designs should be included in the
introduction of the manuscript and the quality factor and mode volume comparison with
these PhC cavities (or related literature survey) should be discussed.
4. The obtained numerical results for the proposed device should be compared to numerical
results obtained by three-dimensional plane wave expansion (3D-PWE) and finite
difference time domain (3D-FDTD) methods.
5. As the authors plot in figures 2(d), 3(d) and 4(d) the variations of quality factors of hybrid
SRR-PCNC structure and coupling efficiency between SRR and PCNC with different SRR
gap widths, relative position and relative angle respectively. It will be better to study the
influence of the SRR gap width, relative position and relative angle on the mode volume of
the proposed resonator.
6. In the fourth section (All integrated on-chip excitation of magnetic dipole mode of
SRR), the authors proposed side-couple the SRR-PCNC structure with a photonic crystal
waveguide, enabling the in-plane accessing geometry. The transmission spectrum of this
structure should be calculated and plotted.
7. Since this is a numerical manuscript, it would be nice if the authors present experimental
results. Right now they just focus on the quality factor of the resonator. Could they perhaps
provide some kind of a more general prediction for how the mode volume of the resonator
depends on the geometrical parameters?

Reviewer 2 Report

The authors presented magnetic dipole modes for plasmonic nanostructures. The topic is worth publishable. My comments are:

1.) The quality and content of the paper need improvement.

2.) Comparison study must be provided with comparing the previous designs.

3.) Recent citations related to the proposed area must be added to keep the readers updated.

4.) Can authors improve coupling efficiency? If not mention the reasons behind it.

5.) Authors are recommended to add the fabricated results to compare with simulated data.

6.) Elaborate Fig. 2-5 in more details.

7.) The conclusion section, to my opinion, should be a little bit enhanced to contain the description of the problem, the methodology and the most important results.

Reviewer 3 Report

In this submission entitled “Exciting Magnetic Dipole Mode of Split-Ring Plasmonic Nano-resonator by Photonic Crystal Nanocavity”, the authors proposed a high-efficient on-chip excitation approach of magnetic dipole (MD) mode of an individual split-ring resonator (SRR) by integrating it onto a photonic crystal nanocavity (PCNC). It is found that the two nano-resonators have very strong mode coupling with an efficiency exceeding 58%. The excited MD mode can be controlled through the gap width of SRR, the relative position and angle between PCNC and SRR. In addition, a photonic crystal waveguide side-coupled with the hybrid SRR-PCNC structure was proposed to excite the MD mode with an in-plane coupling geometry. The authors should however consider and discuss the following points before publication:

  • Besides the metallic SRR, are there any other metallic nanostructures can be used to support the MD mode?   

  • In Fig. 1, what are the Q factors of the EQ and MQ modes of the SRR? Is there a reason why the width of 0.8D is used for the D-shaped air-holes?

  • According to the authors’ discussions, both the permittivity εSRR and permeability μSRR play important roles in the resonant wavelength shift when the SRR gap width is varied. Then, the relationship between the permittivity (permeability) and the SRR gap width should be clearly shown in the manuscript.

  • “Q factor nearly keeps stable if gx > 100 nm.”. Why?

  • “Obvious red shift and sharping of resonant peak are observed from the scattering spectra of hybrid SRR-PCNC structure when the relative angle is changed from 0° to 90°, as shown in Fig. 4(b).”. Please explain this phenomenon in more detail.

  • “The resonant wavelength shift Δλ keeps negative due to negative permeability of SRR and varies periodically with θ (Fig. 4(c)).”. It should be noted that Fig. 4c only shows the resonant wavelength shift as a function of the relative angle. The variations of the permeability of SRR with the relative angle can also be added in Fig. 4c.

  • The obtained excitation efficiency of the MD mode in this manuscript should be compared with other reported schemes.

Reviewer 4 Report

In this contribution, theoretically, a high-efficient on-chip excitation of magnetic dipole model of an individual split ring resonator by integrating it onto a photonic crystal nanocavity is investigated. 

 The manuscript is well organized. The authors may also want to consider the following comments.

  • More detail on simulation such as boundary condition, domain size, and mesh size is needed to add to the manuscript. In addition, what is the surrounding area for SRR? Is it symmetry?

  • In the hybrid structure of SPP and PCNC, the surrounding material is not symmetry one side is covered with siO2, and the substrate is only air. Is this situation considered in the individual simulation for PCNC?

  • Are the air holes in photonic crystals filled with SiO2? Or the SiO2 also has a photonic crystal pattern, as is illustrated in the fig1d. Please explain it correctly in the text.

  • Which simulation tool and condition is used to calculate permittivity and permeability of hybrid and individual structures?

  • Fig.1f1-f2, Which cross-sections?

  • Equation(1) was not found in the ref[26], Please correct it!

Parameters H0 and E0 are not introduced

  • Equation(2) was not found in the ref[28], parameters in eq(2) are not introduced

  • The text could be polished. Some typo problems and mistakes in the text and  introducing the figure
    • Line 106:scattering?
    • Line 134: LC circuit is not introduced.
    • Caption in fig2(d) and (e) are switched
    • Line 159: Fig 2(d)
    • Line 190: parameter XP is not introduced
    • Line 193: fig3(c)

  • Fig1(c1-c4), is not clear which field is presented. In the text, it is introduced as a magnetic field and in the figure caption as an electric field?

  • How dipole moment is indicated in the fig1c3-c4?

Round 2

Reviewer 1 Report

The authors have implemented my suggestions. I recommend this manuscript should be accepted for publication.

Reviewer 2 Report

The paper can be accepted in its present form.

Reviewer 3 Report

The authors revised their manuscript, according to the referee's suggestions.  All the points raised were addressed in detail. I fully recommend publication in Materials in its current state.